# DEML: Drug Synergy and Interaction Prediction Using Ensemble-Based Multi-Task Learning

**DOI:** 10.3390/molecules28020844

**Published:** 2023-01-14

**Authors:** Zhongming Wang, Jiahui Dong, Lianlian Wu, Chong Dai, Jing Wang, Yuqi Wen, Yixin Zhang, Xiaoxi Yang, Song He, Xiaochen Bo

**Affiliations:** 1Academy of Medical Engineering and Translational Medicine, Tianjin University, Tianjin 300072, China; 2Department of Bioinformatics, Institute of Health Service and Transfusion Medicine, Beijing 100850, China; 3Department of Pharmaceutical Sciences, Institute of Radiation Medicine, Beijing 100850, China; 4College of Life Science and Technology, Beijing University of Chemical Technology, Beijing 100029, China; 5School of Medicine, Tsinghua University, Beijing 100084, China

**Keywords:** drug synergy, drug–drug interactions, multi-task learning, ensemble learning, deep learning

## Abstract

Synergistic drug combinations have demonstrated effective therapeutic effects in cancer treatment. Deep learning methods accelerate identification of novel drug combinations by reducing the search space. However, potential adverse drug–drug interactions (DDIs), which may increase the risks for combination therapy, cannot be detected by existing computational synergy prediction methods. We propose DEML, an ensemble-based multi-task neural network, for the simultaneous optimization of five synergy regression prediction tasks, synergy classification, and DDI classification tasks. DEML uses chemical and transcriptomics information as inputs. DEML adapts the novel hybrid ensemble layer structure to construct higher order representation using different perspectives. The task-specific fusion layer of DEML joins representations for each task using a gating mechanism. For the Loewe synergy prediction task, DEML overperforms the state-of-the-art synergy prediction method with an improvement of 7.8% and 13.2% for the root mean squared error and the R2 correlation coefficient. Owing to soft parameter sharing and ensemble learning, DEML alleviates the multi-task learning ‘seesaw effect’ problem and shows no performance loss on other tasks. DEML has a superior ability to predict drug pairs with high confidence and less adverse DDIs. DEML provides a promising way to guideline novel combination therapy strategies for cancer treatment.

## 1. Introduction

In the treatment of complex diseases, such as cancer, monotherapy usually shows resistance due to the biological complexity of the diseases [1,2,3]. Drug combinations have become a promising strategy for solving the problem by improving efficacy and reducing the dose-dependent toxicity [4,5,6]. When the combined effect of two compounds is greater than their single-agent potency, it is considered a synergistic combination. Identifying novel synergistic drug combinations becomes a crucial issue for cancer treatment. However, it is challenging to discover drug combinations by clinical trial and systematic large-scale screening techniques (high-throughput screening, etc.) owing to the vast drug combination search space [7]. Computational methods are urgently required to reduce the search space for drug combinations.

In the past few decades, traditional computational methods have been commonly designed by kinetic models [8], systems biology methods [9], stochastic search algorithms [10], and mathematical methods [11]. These methods are limited by prior knowledge and only accessible to small-scale datasets. With the increase in drug combination data, machine learning (ML) methods provide a more powerful predicting capability [12]. Drug combination prediction algorithms are developed by traditional ML methods, such as Random Forest (RF) [13], support vector machine (SVM) [14], XGBoost and so on [15]. In the recent years, deep learning (DL) methods show superior performance compared to traditional ML. DL methods can handle larger and more complex datasets and have a stronger ability to fit the nonlinear biology process [16,17,18,19,20,21]. For example, DeepSynergy combines cancer cell lines’ untreated gene expression profile with drugs’ molecular information to predict the drug synergy on different cancers [16]. AuDNNsynergy predicts synergy using the multi-omics features [21]. TranSynergy is designed using a self-attention mechanism and explores the extra knowledge through biological networks [18]. MatchMaker constructs two separate sub-networks to extract the drug information as a representation and then concatenates the two representations as inputs into the third feed-forward neural network [17]. MatchMaker achieves the state-of-the-art performance for approaches based on FNN architecture. Deep learning plays an increasing role in accurately predicting drug combinations.

However, combination therapies do not always produce totally beneficial effects as we expect them to. They will result in adverse effects due to pharmacokinetics and pharmacodynamics [4,22,23]. The unexpected effect, termed an adverse drug–drug interaction (DDI), is also one of the major reasons for drug withdrawal [23]. Currently, most previous DL-based studies only focus on predicting drug synergy, while ignoring the prediction for potential adverse DDIs, which may increase the probability of mortality and morbidity. In addition, researchers are still not achieving a consensus on the definition and quantification of synergism [24,25]. There are a variety of synergy quantification models based on different mathematical hypotheses. For example, the most prominent reference model, the Loewe additivity model (Loewe), assumes that drugs that interact with themselves have an additive effect [26]. Another widely used reference model, the Bliss independence model (Bliss), assumes that drug combinations have independent probability distributions [27]. There is also the highest single agent model (HSA) [28], zero interaction potency model (ZIP), and so on [29]. Kuru et al. point out that multiple reference models may give different synergistic labels. Selecting only one certain reference model may introduce error and bias [30]. Thus, drug combination prediction needs to consider comprehensive quantification metrics, and simultaneously to screen the high synergy effect and less adverse DDI candidates for cancer treatment.

Multi-task learning and ensemble learning provides the possibility to solve the above problems. Multi-task learning is a type of transfer learning [31,32]. By learning differences and similarities across tasks, multi-task models can improve both efficiency and quality for each task [33,34], and can also reduce the overfitting risks using other task regularization constraints [35]. However, the main problem of multi-task models is optimization conflicts [36,37], called the ‘seesaw effect’. Ensemble learning combines multiple insights from ensemble structures and can facilitate more accurate prediction. It can reduce the bias and error by shared decision making [38,39,40]. Moreover, it is proven that ensemble structures shared by multiple tasks are capable of alleviating the optimizing conflicts [41,42,43]. 

In this study, we present DEML, which is an ensemble-based multi-task learning approach. DEML can simultaneously optimize seven prediction tasks, including five different synergy regression tasks, one synergy classification task, and one DDI classification task. We combine the cell line transcriptome information and compound chemical information as input features, and train DEML on a large synergy dataset. DEML adopts a novel hybrid ensemble layer to extract multi-perspective higher order representation from input features. The task-specific fusion layer of DEML leverages a gating mechanism to join task-specific information for each task. Finally, the joint information is input into the prediction layer and obtains the prediction for each task. At the main regression task, Loewe synergy score prediction, DEML achieves a root mean squared error (RMSE) of 8.32, a Spearman score of 0.79, and an R2 correlation coefficient of 0.67. At the main synergy classification task, DEML achieves an AUPRC of 0.89, an AUPR of 0.52, and an F1 score of 0.48. DEML outperforms the state-of-the-art drug combination prediction methods on both tasks. In addition, DEML performs better than other widely used multi-task frameworks. Owing to soft parameter sharing and task-specific routing mechanisms, DEML alleviates the ‘seesaw effect’ problem of multi-task learning and shows no performance loss on other tasks. DEML will hopefully screen out synergistic drug combinations with high confidence and less adverse DDIs.

## 2. Results

In this paper, we proposed DEML, which trained on 286,421 samples across 81 cell lines using molecular descriptors and cell line gene expression profiles. DEML can simultaneously predict five synergy regression tasks (Loewe, Bliss, ZIP, HSA, S score), and two classification tasks (drug synergy and DDI classification). DEML is trained in an end-to-end manner. The framework of DEML is shown Figure 1.

The architecture of DEML is composed of three components. The first part is the hybrid ensemble layer (HEL). The HEL is composed of multiple (here six, see Materials and Methods) subnetwork feature extractors. These subnetworks, called experts, have three types with different structures, including dense connection experts, bi-additive experts, and bi-interaction experts. These multiple experts convert the input features into multiple latent vectors. These extracted vectors are used as inputs to the second subsequent module, the task-specific fusion layer (TFL). The TFL consists of different gating networks for each task, respectively, in order to learn task-specific information. The gating networks learn the weight for these multiple latent vectors under the corresponding task using the input feature. Multiple extracted latent vectors obtained by the HEL are weighted and summed into a fusion vector under each corresponding task through the task-specific gating mechanism. Finally, these task-specific fusion vectors are, respectively, input into the corresponding prediction tower for each task of the prediction layer and obtain the corresponding output. These towers are seven parallel double-layer neural networks with the same structure.

We compared the performance of DEML with comparison models using five-fold cross-validation, that is, we shuffled the dataset randomly and split it into five folds. In each cross-validation procedure, three folds were used for training, one for validation (which used for the early-stopping strategy in order to avoid overfitting), and one for test. We iterated five times with a different fold reserved for test purposes each time. Avoiding the prediction deviation introduced by input order, we trained drug pairs both in <x, y> order and <y, x> order. During testing, the outputs of both input orders were averaged as the final prediction. We adopted an early stopping training strategy to avoid overfitting. We saved the training model when we observed the validation loss was not decreased in the following 100 epochs. We averaged the prediction evaluation metrics of the test set under five different data partitions as the final model prediction performance.

We set the Loewe prediction as the main regression task and set the synergy classification as the main classification task which are widely used in drug combination predictions. We compared DEML with several single-task models and multi-task models. The hyperparameters of comparison models were tuned depending on grid search or literature suggestions. We applied the same training strategy applied in DEML for DL-based comparison models to ensure fair evaluation.

### 2.1. Performance Comparison

We compared the performance of the main regression tasks and the main classification tasks, respectively. We mainly compared DEML with the state-of-the-art model MatchMaker. In addition, DeepSynergy, XGBoost, and Random Forest were also considered. In order to achieve the best performance of the comparison model, each single-task comparison model was trained separately in different task. 

For the Loewe prediction task, we evaluated the model performance with three metrics, including RMSE, Spearman, and the R2 coefficients score. The performance result is shown in Figure 2. DEML achieves the lowest test RMSE of 8.32, representing a 14.5% improvement over the second-best performance MatchMaker. DEML reaches 0.79 of Spearman score and 0.67 of R2 correlation coefficient. DEML outperforms all comparison models in terms of both metrics. Through five different data partitions, the model performs stably and the RMSE variance of DEML prediction is 0.0924. DL-based models perform significantly better than ML-based models. 

For the synergy classification task, owing to the imbalance samples (the ratio of positive and negative samples is 1:15), we not only adopted accuracy metrics, but also considered several metrics which are widely used in the evaluation of imbalanced data. The metrics included the AUROC (which tends to be insensitive to the positive-to-negative classes ratio), the AUPR, and F1-score, both of which summarize the performance in terms of recall and precision. The performance is summarized in Figure 3. DEML achieved the largest AUROC of 0.89 for the synergy classification task. MatchMaker achieved the second-largest AUROC of 0.83. Perhaps due to the imbalanced samples and the noise introduced by label definitions, all models showed relatively poor performance for theAUPR and F1-score. However, DEML still achieved a 0.52 AUPR score and a 0.48 F1-score, which were 37% and 42% above MatchMaker, respectively. The performance of all models was indistinguishable using accuracy metrics which may be caused by the sparsity of positive sample space. In addition to accuracy metrics, all models’ performance showed similar trends to the Loewe prediction task.

Overall, DEML achieved the best performance in the Loewe synergy score prediction among the four comparison models. Owing to transfer learning distilling knowledge from other relevant tasks and ensemble mechanisms, DEML outperformed the other comparison models and showed superior improvement over the state-of-the-art model MatchMaker in the synergy classification prediction task. This will be proven in the following section.

### 2.2. Task Ablation Study

We conducted a series of ablation experiments to investigate the impact of different types of tasks transferring knowledge during the performance of the classification task. The knowledge transfer to synergy classification probably comes from two parts, one is the regularization constraint provided by the highly correlated synergy regression task, the other part is the potential knowledge transferred from the DDI classification task. The validation is carried out in two parts.

In order to investigate the contribution of the regression information transfer, we observed the performance from a series of experiments combined with different regression tasks. The results of task ablation experiment are shown in Table 1. Firstly, we defined the DEML which only predicted the synergy classification task as DEML-onlyDDS. Notably, the DEML-onlyDDS single-task model performed slightly better than the MatchMaker single-task model, which might benefit from the ensemble structure. To eliminate the confusion between the improvement caused by ensemble structure and transfer learning, we set the DEML-DDS as the new baseline model. Secondly, we trained the DEML model with five synergy regression tasks and one drug synergy classification task, called DEML-REG. With introducing regression task information, the performance of DEML-REG was significantly improved over the baseline model. DEML-REG outperformed the DEML-DDS by a margin of 0.49% and 0.66% in the AUROC and AUPR, respectively. The synergy classification label is defined by the Loewe score. To investigate whether or not the improvement from regression tasks was mainly depending on the Loewe prediction task, we removed the Loewe prediction task and trained the synergy classification with the remaining four regression tasks, called DEML-removeLoewe. Although the most relevant task information was removed, the performance of the DEML-removeLoewe was similar to the DEML-REG and experienced almost no loss. This result illustrates that the process of task transfer learning is more complex and does not always conform our intuition, for instance, we usually think that two tasks with high-linear correlation can achieve a better knowledge transfer. Perhaps as the Loewe prediction task and other synergy score prediction tasks are all calculated by the same dose-response curve matrix, the DEML is capable of capturing the internal relationship and implicitly learns the information without the Loewe task prediction. 

Next, in order to explore the effect of DDI classification on synergy classification, we proposed a DEML-DDI model. The DEML-DDI model was only trained on drug synergy and the DDI classification task. The DEML-DDI also achieved a significant improvement over the baseline DEML-DDS. The improvement effect is non-inferior to that of the regression task transfer with DEML-REG. This result indicates that there is a complex relationship between the interaction and synergy of drug combinations, as the DDI train task benefits the drug synergy learning capability of DEML. Finally, we found that the performance of DEML trained with all tasks achieves further improvement, which means the knowledge transfer from both regression tasks and DDI task is not redundant.

### 2.3. Expert Ablation Study

DEML adopts the hybrid structure of multiple expert units. The function of each expert unit is to extract the input information of drug x and drug y, and then fusion the information of the two drugs and convert it into a latent vector. The main difference between the three expert units is the interaction pattern of the fusion process. For example, the bi-additive expert unit extracts the information of two drugs, respectively, and performs the addition operation for the two drugs to achieve the information interaction. It indicates the relationship between the information of two drugs using an ‘or’ interaction pattern. We propose a novel bi-interaction expert that uses the bitwise product method to interact the two information vectors. It indicates the relationship between the information of two drugs using an ‘and’ interaction pattern. This pattern is proven to strengthen the interaction capability, which is widely used in factorization machines (FM) [44], neural factorization machines (NFM) [45], KGAT [46,47] and so on. DEML combines the multiple experts aims to extract the drug interaction information using different perspectives. We performed an expert ablation experiment to prove whether the hybrid expert structure has a positive impact on model prediction.

We designed multiple different combinations using three candidate experts and evaluated the prediction performance. Considering the size of the network and the prediction performance of the model, we set the sum of expert units to six for each experiment, for example, when two types of expert structures are adopted, each number of experts is set to three. When one type of expert unit is adopted, the number of this expert unit is set to six. The fixed sum number of experts aims to compare the performance of the model fairly. We observed that when the HEL only adopts one type of expert structure, the bi-additive expert unit and the bi-interaction expert unit have a significant advantage over the dense connection expert unit. The DEML composed of bi-interaction expert cells performs best of these three. Although the use of the bi-interaction expert unit alone has achieved a high prediction performance, DEML achieves the highest prediction performance when multiple types of expert units are adopted at the same time. It seems DEML represents the fusion information of two drugs from multiple perspectives, reducing the redundancy of experts of the same type. The results of the expert ablation experiments are shown in Table 2.

### 2.4. Seesaw Effect Study

The ‘seesaw effect’ is a major issue in multi-task learning, especially for different tasks with complicated correlations. This means the performance of some tasks is often improved at the sacrifice of the decreased performance for other tasks. The ‘seesaw effect’ is mainly caused by two reasons. First, multi-task learning may suffer from optimization conflicts, as the parameters are extensively shared by all tasks when using hard sharing structures. Second, many multi-task models are not good at learning the differences across all tasks, which is caused by the lack of task-specific knowledge distilling modules. The performance will be further decreased when there are little correlations between different tasks, such as the regression and classification synergy tasks, and the DDI and synergy prediction tasks. We adopted the following two strategies to address the above issues. In order to achieve more flexible parameter sharing, DEML reduces dependency on the same parameter from different tasks using a multiple expert structure. DEML strengthens the task-specific learning ability through a task-specific gating mechanism. 

DEML exhibits a superior improvement in the Loewe prediction task and synergy classification task. In order to investigate whether the DEML model on the remaining tasks shows performance loss, we next compared the performance of DEML for the remaining tasks with several single-task models. We adopted MatchMaker and DeepSynergy as comparison models, which showed excellent performance in the previous section. For all the single-task models, we trained each task separately in order to obtain the best performance from comparison models. Figure 4 shows the result. DEML achieves the best performance on all evaluate metrics in the four regression tasks, including ZIP, Bliss, HSA and S score prediction. DEML shows better performance among the four regression tasks when predicting HSA score. The same trend is observed when using MatchMaker and DeepSynergy. This result is consistent with the high correlation between the HSA score and the Loewe score (See Appendix A). It seems more relevant tasks show a more similar performance. As the DDI prediction task is easy to learn, all prediction values and real values of DL-based methods are in good agreement. Both DEML and MatchMaker achieve an AUROC and AUPR exceeding 0.99, and DeepSynergy also shows an AUROC and AUPR close to 0.99. DEML outperforms the other single-task models in the other synergy score prediction tasks and shows no performance loss in the DDI prediction tasks.

In addition, to investigate the impact of task-specific gating components and whether DEML can better alleviate the ‘seesaw effect’ issue, we compared our model with several existing multi-task frameworks which are widely used. First, we constructed the share-bottom (SB) model [31,48]. The SB model shares the same bottom extraction structure across all tasks and puts the extraction representation into task-specific towers. The SB model performs better when there are higher correlations across different tasks. SNR-trans and SNR-aver are proposed by Jiaqi Ma [41], which use sub-network routing mechanisms to strengthen the resolving ability of task-common and task-specific information. The major difference between the two SNR models is that SNR-trans uses a transformation matrix to convert the low-level representation to high-level representation and SNR-aver does not. These alternative multi-task models apply the same training strategy as DEML. 

The performance of multi-task models for all synergy scores’ prediction tasks is summarized by Figure 5. All the comparison models achieve the highest correlation coefficients score among the five tasks when predicting the Loewe score. Except fromt he SNR model, the other multi-task models outperform the state-of-the-art single model in the Loewe prediction task. The SB model, which adopts the same bottom structure as DeepSynergy, achieves a higher Loewe prediction correlation coefficient score than single-task DeepSynergy, and even outperforms MatchMaker. This proves that most multi-task frameworks can provide additional improvement in the Loewe prediction from the relevant tasks. However, including the SB model, all the comparison models show different levels of performance loss in the other regression prediction tasks than the single-task models. We find both SNR-trans and SNR-aver even perform worse than other comparison models. One possible reason for this is that the complex routing mechanism designed by binary coding variables is hard to train and optimize. Furthermore, we also find that the R2 score of the SB model for the S score prediction task decreases by roughly 49% compared with the DeepSynergy single-task model, which exhibits the most severe performance loss among the five synergy score prediction tasks. A similar trend is observed in other comparison models. Conversely, the HSA prediction task usually experiences the lowest performance loss. This is likely caused by the HSA score which maintains a high correlation with the Loewe score and the S score shows a poor correlation. Additionally, we compare the DEML-onegate model in order to analyze the contribution of the task-specific gating mechanism. In the DEML-onegate model, the multiple task-specific gating of the DEML model was replaced by one task-common gating. Though DEML-onegate achieves better performance than the other comparison models, DEML-onegate experiences slight performance loss compared with the MatchMaker model in some tasks. Combined with the task-specific gating mechanism, DEML outperforms DEML-onegate across all regression tasks.

### 2.5. Feature Importance Analysis

In order to better understand the relationship between gene expression and the synergistic effects of drug combinations, we analyze the feature importance. We adopt the Deep SHAP algorithm and calculate the approximate Shapley values [49] to represent the features importance values. The Shapley value is a famous measurement which provides the marginal contributions of each component. This value can indicate how input feature values result in the predicted synergy task. Deep SHAP [50] is an enhanced version of the DeepLIFT [50] algorithm and is compatible with deep learning models. 

Using 1000 samples selected randomly as background samples, we approximate the conditional expectations of Shapley values. In the chemical descriptor features, the variance between this feature of some dimensions above all sample sets is zero, which means that the features of these dimensions have not changed under different samples, and these features have been removed.

Here, we mainly analyze the influence of features of the synergy classification task in three aspects, including contribution analysis of different types of features, biological analysis of key features, and key genes of specific cell lines.
(a)Contribution analysis of different types of features.

Our model uses chemical information and gene expression profile information to predict synergy classification tasks. For the chemical descriptor features of drug x and drug y, 424 features of 541 remained, and the Shapley scores of the two drugs under these features were averaged as the final importance scores of the chemical descriptor. In addition, we also analyzed the importance score of the dimensions of 927 gene expression features. We visualized the importance values of all features at Figure 6a. We also analyzed the contribution of two types of features and show the results in Figure 6b. The contribution is calculated by summing the whole feature importance values of each type. The contribution of gene expression feature accounts for 61.6%, which is higher than the contribution of drug descriptors. We also calculated the average importance values of all features, represented by a red line in Figure 6a. The average importance value is 0.017. We found 296 importance values of genes higher than the average value, and there are 243 drug descriptor features. Gene expression profile information contains more important information in synergy classification tasks, not only as the function of identifier differentiation.
(b)Biological analysis of key features.

We further analyzed the 286 importance genes (the gene with an importance value higher than the average value). We conducted a global analysis of the importance genes, including two types of gene enrichment analyses, a KEGG pathway, and a Gene Ontology Biological Process. We found that these importance genes were significantly enriched in 65 KEGG pathways and 366 GOBPs (adjusted *p*-value < 0.01). Figure 6c,d shows the top 20 enrichment results. For the enrichment result of GOBPs, an intrinsic apoptotic signaling pathway shows the highest scores. Moreover, there also exists a part of biological process that is apoptotic. For the enrichment result of KEGG, most of the pathway is relevant in cancer and the regular process of cancer.
(c)Key genes of specific cell lines.

Finally, we further analyzed the cell-line-specific key genes in different cancer types. We averaged the Shapley values in the cell-line-specific dataset as the importance value of gene expression features under certain cell lines. We selected four representative cell lines with a large sample size, including MCF7, A549, HT29, and Sk-OV-3. Then we obtained the top 10 genes with higher importance values as the key genes in the different cell lines. The results are shown in Table 3. A key role may be played by the top 10 genes involved in cancer cell lines. For example, the PGM1 of key genes under the MCF7 cell line, is proven to be associated with breast cancer [51]. The TSPAN14 of key genes under the A549 cell line, is found to be a potential biomarker and provides a theoretical basis for the pathogenesis of lung adenocarcinoma [52].

### 2.6. Consistency Analysis of Synergistic Samples

According to the previous analysis, the Loewe score has excellent ability in distinguishing synergistic and non-synergistic (including additive and antagonistic) drug combinations. Since these five synergy scores are based on different biological assumptions, only considering a single synergy reference score may introduce bias. Zagidullin et al. suggests that drug combinations will be indicated as more confident synergistic candidates when achieving high scores in more synergy scores [24]. Thus, it is necessary to consider the consistency of multiple synergy scores in synergistic samples. 

In order to investigate the confident predictive capability of classification tasks in synergistic samples, we then analyzed the relationship of the predictive label with different synergy scores.

We visually inspected the correlation of pairwise synergy scores and the synergistic sample distribution in Figure 7. We selected the model with the highest prediction performance as the final prediction model and extracted the results from the test set. The top right section of the figure shows the correlation between pairwise synergy scores and the point in each subfigure represents a sample with a pair of synergy scores. The bottom left subfigures show the Pearson score of the pairwise synergy score, the red numbers represent the Pearson scores on positive (synergistic) samples, and the black numbers represent the Pearson scores on all test data. Figure 7a shows the distribution of synergistic points defined by the ground truth Loewe score using threshold values. A number of ground truth synergistic points (yellow) appear in the brown box where the pair of synergy scores are inconsistent. This means that one synergy score indicates high synergy and the other does not. However, Figure 7b shows the synergistic nodes predicted by the model (yellow) shows higher consistency and these are hardly found in the green box. The green box position is the same as the brown box. Most Pearson scores of pairwise synergy scores on predictive synergistic points are higher than that of ground truth synergistic points. This implies that under the predictive synergistic samples, two synergy scores have a higher correlation. The DEML model tends to judge the sample as a positive sample when multiple types of synergy score are high at the same time.

### 2.7. Prediction Deviation

The correctness of the label in DrugComb would also be interrupted by systematic bias or experimental noise. These noisy labels will deteriorate the performance of synergy classification tasks and introduce predictive deviation. 

Based on the analysis in the previous section, DEML shows the ability to tend to predict more confident positive samples. We then further checked whether the DEML had the potential ability to correct the noisy label or not, that is, the model predicts the combination as synergism while the combination is incorrectly labeled as non-synergism. We collected the drug combination samples with high synergy predictive probability exceeding 0.99, totaling 167 samples. The ground truth label refers to the actual synergy label defined by the actual Loewe score. Among these 167 samples, there were 25 drug combinations that were inconsistent with the ground truth label (meaning the actual Loewe score was below five). In order to make the predictive samples more reliable, we selected 18 of 25 drug combinations also showing high prediction values under the other synergy measurement scores, including 15 drug combinations with the five prediction values higher than five and 3 combinations with prediction values exceeding five in terms of four predictive scores. These samples are given in Table 4. They were highly confident to be predicted as positive samples but were labeled as negative samples, which was probably caused by incorrect labeling. We checked the other four actual synergy scores (actual Bliss, HSA, ZIP, S score) of these 18 combinations to determine the incorrect label. In Table 4, there are 4 of 18 combinations showing higher actual synergy scores above five in terms of all of the remaining synergy measurement scores. There exist another five combinations exceeding five under three actual synergy scores. Note that if we liberalize the threshold to 0, 14 of 18 show as synergistic in terms of more than three actual synergy scores.

We also discovered several verifications and some experimental results reported by the literature which prove that a part of above samples which show synergy effects is incorrectly labeled as non-synergistic, for example, the combination of Gefitinib and Dasatinib are reported to achieve a better response rate and duration for lung cancer treatment [53]. Vemurafenib and Thalidomide are used together to treat cancer [54]. Zolinza is one of the HDAC inhibitors. MK-2206 shows synergistic effects in combination with HDAC inhibitors. It further confirms the accuracy of the DEML prediction [55].

### 2.8. Application

We further applied DEML to predict the more confident synergistic drug combinations and we screened out 15 drug combinations on each specific cell line. These samples obtained special high predictive synergy score in all types of synergy measurement (greater than 30). The prediction scores are almost in agreement with the actual synergy score. Multiple high actual synergy scores indicate that the drug combination has a high possibility of synergistic effects. The results are shown in Table 5. We also discovered the validation results of the literature in some drug pairs.

For example, Temozolomide and Salubrinal show synergistic effects in glioblastoma cells [56] and Salubrinal is capable of upregulating the expression of ATF4 and enhancing apoptosis induced by Temozolomide. Vemurafenib can increase the serum concentration of Actinomycin D [57]. Combining them has the potential to enhance the therapeutic effect. 

We obtained 167 pairs of drugs with high synergy probability as predicted by the model. The network predicts that 14 of the 167 have adverse DDIs which we should avoid. These drug combinations are shown in Table 6.

For example, in these 14 pairs of drug combinations, the combination of Lapatinib and Docetaxel could increase gastrointestinal toxicity [58]. Combining Sunitinib with Docetaxel could raise several safety issues, such as neutropenia, skin disorders, constitutional disorders, and so on. The combination of Erlotinib and Dasatinib can induce skeletal muscle toxicity [59]. The risk of treatment will be increased if these combination therapies are adopted without careful consideration.

DEML can provide more confident synergy drug pairs using multiple synergy score prediction to avoid the potential adverse DDIs.

## 3. Conclusions and Discussion

In this study, we developed a novel multi-task deep learning method, DEML, which can predict five different synergy scores and two classification labels (including synergy classification predictions and DDI predictions). Leveraging transfer learning from each relevant task and the ensemble expert structure, DEML achieves the RMSE of 8.24, Spearman score of 0.79, R2 score of 0.67 in the Loewe prediction task, and achieves an 0.89 AUROC, AUPR, and F1 score, outperforming the comparison models. Using the task-specific gating mechanism, DEML alleviates the ‘seesaw effect’ issue and outperforms several widely used multi-task models. Further, we analyzed the feature importance of different features and obtained the important biological processes and key genes produced by the model. We analyzed the relationship between different synergy measure scores and discovered that the predictive synergistic samples tend to show high consistency under multiple synergy scores. Moreover, DEML is capable of screening out the synergy candidates with high synergy scores in all synergy measurements and avoids potential DDIs.

In our study, we simply labeled the drug synergistic or non-synergistic using the Loewe score with threshold values. DEML can constrain the prediction with higher confidence by optimizing other multiple synergy scores. The label noise introduced by systematic bias cannot be totally eliminated and causes prediction deviation. During DEML training, we used smooth label cross-entropy loss to alleviate the problem. In future work, more noise-robust strategies need to be adopted to solve the noisy label adverse effect. Moreover, a more sensitive synergistic evaluation measurement needs to be established, which can better distinguish between positive and negative samples. 

Ensemble expert structures and multi-task transfer learning play a key role in prediction performance. In our study, we simply adopt three different types of feed-forward neural network structures as experts to extract the interaction information, which is easy to train. However, in recent years, graph neural network (GNN)-based methods have shown excellent prediction performance and perform better to extract the information from molecular graph structures end to end. There are already several GNN-based models showing higher accuracy in drug synergy prediction. Through appropriate training strategies and GNN-based structural improvement, DEML can hopefully achieve better performance.

Synergy classification prediction tasks usually suffer from the issue of unbalanced data. It is hard to distinguish the synergy samples in the sparsity positive sample space. Positive sample prediction is the focus of our study. There is still much room for improvement in AUPR and F1 metrics. Tree models have great potential here, as they are both interpretable and capable of handling unbalanced data. In recent years, tree models have made progress in predicting drug combinations. Wu et al. proposed an enhanced deep forest method which can alleviate the unbalanced data problem [51]. This may suggest that improving the structure and adopting an unbalanced strategy, such as positive data argument, can alleviate the issue.

## 4. Materials and Methods

### 4.1. Data

*Label data*. In this study, synergy effects of drug pairs were obtained from the DrugComb dataset, which contains 466,033 drug combinations under 112 cell lines. DrugComb integrates and standardizes these drug combination dose–response data from 37 studies, mainly from NCI Almanac study. The data are available from https://drugcomb.fimm.fi/, accessed on 1 December 2022 (version v1.4). Since some drug combinations lack feature information, 335,692 combinations under 81 cell lines remained after feature alignment. The synergy score was reported by only 1 experiment in 81% of combinations, while it was reported by more than 1 experiment (2 to 8) in the remaining combinations. We averaged the synergy score of replicate experiments. Finally, the processed data contained 286,421 drug combinations under 81 cell lines, which included 3040 drugs. During the multi-task prediction, 3 types of labels were considered.

(a) Synergy label for regression prediction. 

We collected 5 commonly used synergy scores, including Loewe, Bliss, HSA, ZIP, and a novel synergy score termed S score. All synergy scores were provided by DrugComb. The definitions of these synergy scores are shown in Appendix A. Then we analyzed the correlation of different synergy scores. Correlations between 2 different synergy measurement scores were not as strong as expected (Appendix A). For some drug-pairs, 1 reference measurement provided a high synergy score while the other provided a low synergy score, which is inconsistent. Bliss and HSA achieved the highest Spearman score, 0.727, and the Loewe and S score achieved the lowest Spearman score, 0.094. Only select single certain synergy scores may have introduced bias. Thus, we predicted the 5 synergy scores simultaneously in order to obtain a comprehensive synergy measurement to evaluate drug synergy. The distributions of 5 synergy scores are shown in Figure 8.

(b) Synergy label for classification predictions. 

The widely used method in synergistic classification studies is using a threshold value of synergy scores to distinguish between synergistic and non-synergistic effects. The median of the Loewe synergy score was −4.59 and the mean value was −8.69 (Appendix A). In this study, in order to rigorously screen samples with high synergistic potential, the threshold was set to 5. Samples with synergy scores greater than 5 were considered to be positive synergistic samples and those less than 5 were considered to be negative samples.

(c) DDI labels for classification prediction. 

The DDI label information used in this study were obtained from the Drugbank (v5.12) database. The processed drug pairs were classified into 18 interaction relationships according to Drugbank’s description. A total of 16 out of 18 involved adverse DDIs, such as increased risk of cardiotoxicity, and only a few samples were non-harmful DDIs. We classified the adverse DDI drug pairs as positive samples which we wanted to avoid, and classified the other drug pairs as negative samples, which included non-harmful DDI types and drug pairs not recorded by the Drugbank database. The distribution of DDI labels is shown in Appendix A. The label information for both classification tasks is summarized in Table 7.

*Feature data*. Two types of features were used when training the model: normalized gene expression profile features and chemical descriptor features. Normalized gene expression profile features, which were obtained from Iorio’s study, provided the specific untreated cell line information. In order to reduce the feature dimension, we chose 927 landmark gene expression features as final cell line features. These landmark genes were obtained from Subramanian’s study. Chemical descriptor features, which were calculated by ChemoPy tools, provided the chemical drug information. The chemical descriptor is a 541-dimensional vector representation. Each dimension presents the physico-chemical properties. The feature information is summarized in Table 8.

### 4.2. The DEML Architecture

DEML is an ensemble-enhanced multi-task deep learning model for 5 regression tasks (predicting 5 different drug synergy scores), and 2 classification tasks (synergy classification and DDI classification). The whole architecture is shown Figure 1. DEML contains 3 major components. Firstly, the hybrid ensemble layer extracts the input features into multiple higher-order vectors using different perspectives. Secondly, the task-specific fusion layer learns task-specific weights and sums the weighted vectors based on each task separately. Finally, each summing vector is input into corresponding prediction towers to obtain the task-specific prediction output. The entire workflow is shown in Figure 1.

*Hybrid Ensemble Layer*. The bottom neural network structure is capable of automatically converting input information into high-order information. The conventional multi-task model uses a hard parameter sharing the bottom structure. The parameters learned at the bottom are shared by all tasks, which suffer from optimization conflicts. This means that the optimization of one task may compromise the performance of other tasks.

We proposed a novel hybrid ensemble layer (HEL). The HEL leverages a soft parameter sharing mechanism to alleviate the strong dependency of multiple tasks on the bottom structure parameters, which can more profitably optimize multiple targets. The HEL consists of a group of sub-networks. These sub-networks, called experts, have N different types. Each type of expert adopts distinct neural network structure to achieve different interaction information extraction patterns of drug x and drug y combinations, which reduce information redundancy caused by the same interaction pattern. The number of experts of each type can be single or multiple. Each expert coverts the input features into the d-dimension latent vector. In this paper, we mainly adopt 3 types of experts as follows.
Dense connection expert. For the drug combination, x, y, $d_{x}$ and $d_{y}$ refer to the molecular representation of drug x and drug y, respectively. $c_{r}$ is the cell line gene expression profile feature. The dense connection expert’s output $O_{Dense}$ is defined as Formula 1.


(1)
$$O_{\mathrm{Dense}}{=\mathrm{Layer}}^{\left(2\right)}\left({\mathrm{Layer}}^{\left(1\right)}\left(\mathrm{CAT}\left(d_{x}{,d}_{y}{,c}_{r}\right)\right)\right),$$



(2)
$${\mathrm{Layer}}^{\left(n\right)}\left(x\right)=\mathrm{Dropout}\left(\mathrm{BatchNorm}\left(\mathrm{ReLU}\left({\mathrm{Linear}}^{\left(n\right)}\left(x\right)\right)\right)\right),$$


CAT represents the concatenation of multiple input features. ${\mathrm{Layer}}^{\left(n\right)}$ refers to the *n*-th layer of the neural network. Linear represents a linear neural network. In brief, the dense connection expert consists of 2 neural network layers. Most of the neural network layer of DEML consists of rectified linear unit (ReLU) activation, batch normalization, and the dropout layer. The dropout operation is capable of improving the robustness of the model and avoiding dependence on a single neural unit. It can alleviate the overfitting problem. The dropout probability is set to 0.5.
2.Bi-additive expert. The molecular representations of drug x and drug y are, respectively, concatenate with gene expression profile features, and then 2 concatenated features are input into the double tower structure. Both towers have a symmetrical structure and are composed of 2-layer neural networks. ${\mathrm{Tower}}_{x}$ coverts the information of drug x into vector x, ${\mathrm{Tower}}_{y}$ converts the information of drug y into vector y. Both vectors are added bitwise to obtain the bi-additive expert output $O_{BI-additive}$.
(3)$$O_{BI-additive}={\mathrm{Tower}}_{x}\left(\mathrm{CAT}\left(d_{x},c_{r}\right)\right)+{\mathrm{Tower}}_{y}\left(\mathrm{CAT}\left(d_{y},c_{r}\right)\right),$$
(4)$$\mathrm{Tower}\left(x\right)={\mathrm{Layer}}^{\left(2\right)}\left({\mathrm{Layer}}^{\left(1\right)}\left(x\right)\right),$$3.Bi-interaction expert. As a commonly used double-tower based model, NFM applies a bi-interaction structure to enhance the ability to cross feature information. In this paper, the structure is simply improved to form a bi-interaction unit. The features are input into a double tower which is similar to the bi-additive tower, resulting in 2 output vectors. These 2 vectors are then added bitwise to obtain the vector $O_{add}$. In addition, the 2 vectors are produced bitwise to obtain the vector $O_{product}$. $O_{add}$ and $O_{product}$ are transformed by different single-layer neural networks and added bitwise again to obtain the output $O_{BI-interaction}$. The single-layer neural network is represented by ${\mathrm{Layer}}_{add}$ and ${\mathrm{Layer}}_{product}$. The symbol ⨀ indicates product operation. The $O_{BI-interaction}$ is computed as follows:(5)$$O_{add}={\mathrm{Layer}}_{add}\left({\mathrm{Tower}}_{x}\left(\mathrm{CAT}\left(d_{x},c_{r}\right)\right)+{\mathrm{Tower}}_{y}\left(\mathrm{CAT}\left(d_{y},c_{r}\right)\right)\right),$$
(6)$$O_{product}={\mathrm{Layer}}_{product}\left({\mathrm{Tower}}_{x}\left(\mathrm{CAT}\left(d_{x},c_{r}\right)\right)⨀{\mathrm{Tower}}_{y}\left(\mathrm{CAT}\left(d_{y},c_{r}\right)\right)\right),$$
(7)$$O_{BI-interaction}=O_{add}+O_{product},$$

*Task-specific fusion layer*. The HEL contains M experts, which produces M latent vectors. Multiple vectors represent drug interaction information from different perspectives. We proposed a task-specific fusion layer (TFL), which contains multiple separate gating units. Gating units learn the weight for each latent vector through the features of drug combination and cancer cell lines. In order to avoid the performance degradation caused by sharing the same gate parameters for multiple tasks, TFL adopts K different gating units for each task. Thus, K is equal to the number of tasks. TFL can automatically learn to select the relevant information from multiple hidden vectors for each task.
(8)$$W_{k}=\mathrm{softmax}\left({\mathrm{gate}}_{k}\left(CAT\left(d_{x},d_{y},c_{r}\right)\right)\right),$$(9)$${\mathrm{gate}}_{k}\left(x\right)={\mathrm{Layer}}_{k}^{\left(2\right)}\left({\mathrm{Layer}}_{k}^{\left(1\right)}\left(x\right)\right),$$

Gate represents the gating units for task k. It is a 2-layer neural network. $W_{k}$ is the M-dimensional weights obtained through softmax conversion of the output of the gating units. $I^{num}$ refers to the output vector from the *num* expert. The final fusion representation for task k is formulated as:(10)$$O_{k}=\sum_{num=1}^{M}W_{k}^{num}\times I^{num},$$

*Prediction layer*. The prediction layer consists of 2 types of prediction towers, called the regression tower and the classification tower. It should be noted that the prediction tower is slightly different from the previous tower structure of the bi-additive expert. The final layer is simply a linear layer. Using the fusion representation $O_{k}$ as the input, the prediction tower for task k predicts the $y_{k}$ for certain tasks. The output of each regression tower is a 1-dimensional vector representing the synergy score. The output of the classification tower is a 2-dimensional vector, and the vector is then fed into a softmax layer to compute the prediction probability of positive and negative samples, respectively. The category with the highest probability is taken as the classification label.
(11)$$y_{k}={\mathrm{Linear}}_{k}\left({\mathrm{Layer}}_{k}\left(O_{k}\right)\right),$$

### 4.3. Compared Methods

*Single-task model*. We compared the DEML with 4 single task models which showed excellent prediction performance in the synergistic drug combination prediction tasks, including 2 DL methods and 2 ML methods. (1) DeepSynergy, which was proposed by Preuer K. This algorithm is based on multi-layered deep neural networks. (2) MatchMaker, which was proposed by Halil Ibrahim Kuru. Different from DeepSynergy, this algorithm adopts 2 drug-specific subnetworks and 1 synergy prediction subnetwork. In this method, the embeddings of the drug combination are first learned separately and then are concatenated to predict the synergy score. (3) Random Forest, which was adopted to predict drug synergy, is an ensemble ML method consisting of multiple decision trees. (4) XGBoost, which is an ensemble ML method, is based on the gradient boosting mechanism and parallel promotion.

Since the size of the training set was similar to the study of MatchMaker and DeepSynergy, we used the same hyperparameter settings for the comparison DL methods discussed in this study. The neuron number of DeepSynergy is set to {2048, 1024}. The drug specific subnetworks of MatchMaker consist of double-layer neural networks with {2048, 1024} neurons. The synergy prediction subnetwork consists of double-layer neural network with {1024, 512}.

The hyperparameters of the remaining ML methods, including Random Forest and XGBoost, are tuned by grid search. The search range of hyperparameters can be found in Appendix A. The implementation of these ML methods involves scikit-learn tools. 

*Multi-task model*. To go further, we compared DEML with several multi-task DL frameworks which are widely used, including the share-bottom model and 2 variants of the SNR model. (a) The share-bottom model, which was proposed by Caruana, is frequently used in the biomedical field. This model consists of a bottom structure and a prediction tower. All tasks share the same bottom structure parameters. To guarantee a fair comparison, the bottom structure is constructed to be the same as the bottom structure of the comparison model, DeepSynergy. (b) SNR-trans, which was proposed by a Google research group, plays an important role in business recommended system applications [41]. The SNR model applies sub-network routing mechanisms to achieve flexible parameter sharing. The SNR-trans model in this study contains 2 layers. Each layer consists of 6 subnetworks, which are formed by single-layer neural networks with 1024 neurons. A learnable binary variable, called a coding variable, controls the connection of subnetworks. (c) SNR-aver, which is a simplified version of SNR-trans. SNR-aver removed the transform matrix between subnetworks. The remaining settings of SNR-aver are similar to SNR-trans. Both prediction towers of these multiple-task comparison models are the same as the DEML tower structure.

### 4.4. Model Training

During DEML model training, a pair of drug chemical features and cell line genomic features were provided as the input. These features, concatenated by a special pattern, were input into 3 types of 6 experts. The dense connection expert is a double-layer network with {4096, 2048} neurons. The tower structure of the bi-additive expert and bi-interaction expert is also a double-layer network with {4096, 2048} neurons. The output latent vectors of experts were input into the task tower through the task-specific gating mechanism weighted sum. The neuron number of each gating network is set to {4096, 7}. The neuron number of the prediction tower is set to {2048, 2048}.

The output of DEML consists of classification prediction and regression prediction. The training loss of the regression task is MSE between prediction and ground truth. Specially, we used smooth cross-entropy (SCE) loss instead of cross-entropy (CE) loss for classification prediction tasks. It is proven that SCE performs better in classification conditions with noisy labels. ε is the hyperparameter indicating the degree of label smoothing. SCE loss is calculated as:(12)$$\mathrm{CE}\;\mathrm{Loss}=-\sum_{i=1}^{m}p_{i}\log q_{i},$$
(13)$$\mathrm{SCE}\;\mathrm{Loss}=\left\{\begin{array}{c} \left(1-\varepsilon \right)\times \mathrm{CE}\;\mathrm{Loss},\;if\;\left(i=y\right)\\ \varepsilon \times \mathrm{CE}\;\mathrm{Loss},\;if\;\left(i\neq y\right) \end{array}\right.,$$

The prediction value of classification tasks varied with a range of 0–1, the scale of which is much smaller than the distribution of regression prediction values. We allocated a weight of 10 for classification loss and a weight of 1 for regression loss in order to balance these 2 contributions. The weighted sum of all task losses was taken as the final objective of model optimization. We optimized the objective function using ADAM with the learning rate 0.0001 and batch size 128. We initialized all model parameters using Xavier methods.

### 4.5. Evaluation Metrics

We trained the models with regression tasks and classification tasks. We evaluated the performance of 2 types of tasks with different metrics.

For regression task evaluation, we employed the root square error (RMSE), Pearson, and Spearman correlation coefficient as metrics. We defined $Y_{i}$ as the actual synergy score of the sample *i* and $y_{i}$ as the prediction score. $\bar{Y}$ and $\bar{y}$ were the average of them. The m refers to the size of the test set used to evaluate the performance of the model. The definitions of these metrics are as follows:(14)$$\mathrm{RMSE}=\sqrt{MSE},$$
(15)$$\mathrm{Spearman}=1-\frac{6\sum_{i=1}^{m}d_{i}^{2}}{m\left(m^{2}-1\right)},$$
(16)$$R^{2}=1-\frac{\sum_{i=1}^{m}{\left(Y_{i}-y_{i}\right)}^{2}}{\sum_{i=1}^{m}{\left(y_{i}-\bar{Y}\right)}^{2}},$$

For classification task evaluation, we employed the accuracy F1 score to access the model performance. In addition, we employed the AUROC, which refers to the area under the operating characteristic (ROC) curve, and the AUPR, which refers to the area under the precision-recall (PR) curve as metrics. The definitions of accuracy and F1 score are as follows. *TP*, *TN*, *FP*, and *FN* refer to true positive, true negative, false positive, and false negative, respectively.
(17)$$\mathrm{Accuracy}=\frac{TP+TN}{TP+TN+FP+FN},$$
(18)$$\mathrm{Precision}=\frac{TP}{TP+FP},$$
(19)$$\mathrm{Recall}=\frac{TP}{TP+FN},$$
(20)$$F1\;\mathrm{score}=\frac{2\times \mathrm{Precision}\times \mathrm{Recall}}{\mathrm{Precision}+\mathrm{Recall}},$$

## Figures and Tables

**Figure 1 molecules-28-00844-f001:**
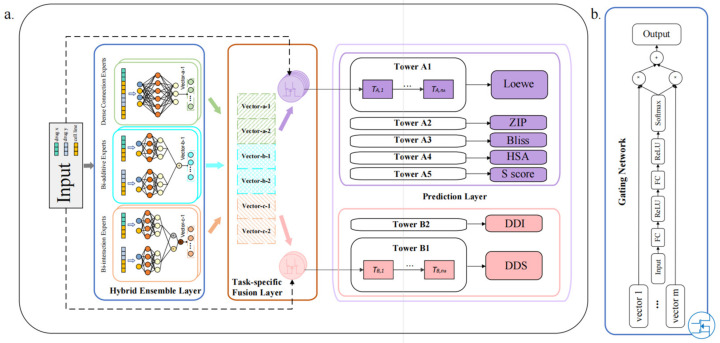
(**a**) DEML architecture. Hybrid ensemble layer (HEL). The feature of drug combination with the certain cell line is input into the HEL. The HEL is an ensemble structure and consists of different types of expert subnetworks. Each expert converts the input feature into a hidden vector, such as the vector-a-1 in the figure. Task-specific fusion layer (TFL). The TFL fusions the multiple output vectors of the HEL and weights them into representations within different task-specific gating networks. Prediction layer. Each tower takes the task-specific representation to obtain the prediction for each task. The purple box represents the regression tower and the pink box represents the classification tower. Note that DDS refers to the drug synergy classification prediction. (**b**) The structure of the gating network. The gating network can learn the weights using input features automatically.

**Figure 2 molecules-28-00844-f002:**
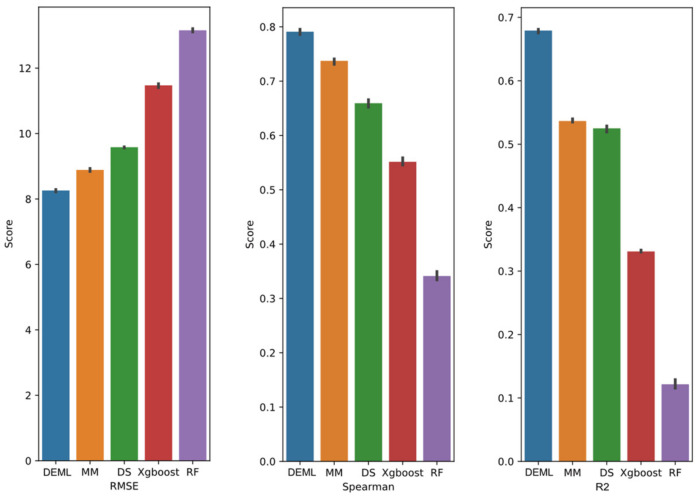
Regression prediction performance of DEML and single-task comparison algorithms in Loewe regression prediction task. The error bar represents 95% confidence interval of five-fold performance.

**Figure 3 molecules-28-00844-f003:**
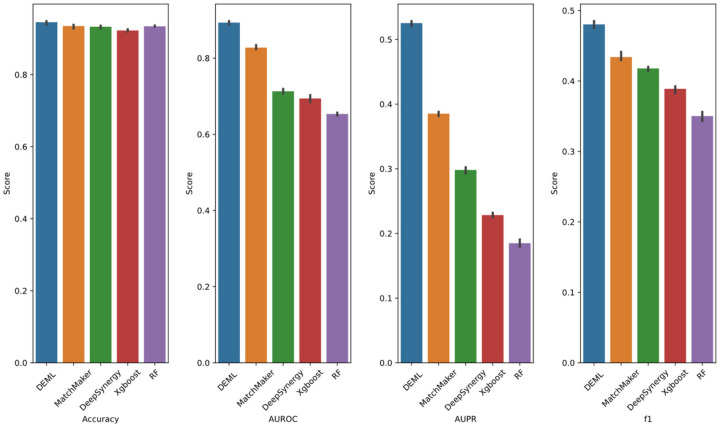
Classification prediction performance of DEML and single-task comparison algorithms in synergy classification prediction task. The error bar represents 95% confidence interval of five-fold performance.

**Figure 4 molecules-28-00844-f004:**
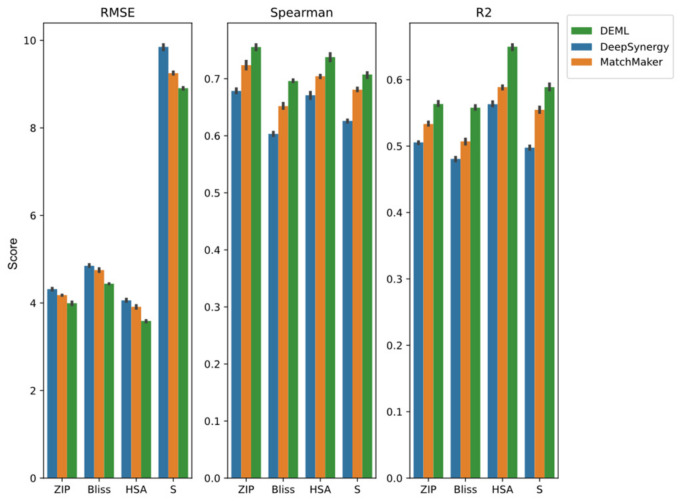
Prediction performance of DEML and single-task comparison algorithms in other four regression prediction tasks. The error bar represents 95% confidence interval of five-fold performance.

**Figure 5 molecules-28-00844-f005:**
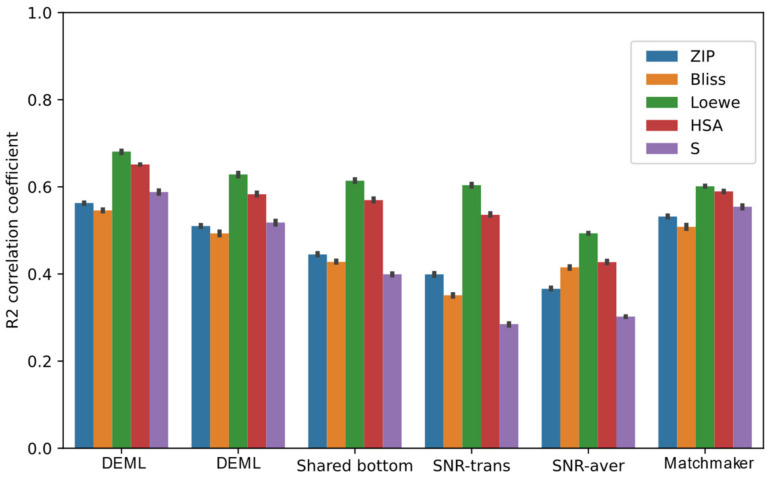
Prediction performance of DEML and multi-task comparison algorithms in all synergy regression prediction tasks. MatchMaker single-task refers to the best prediction performance with MatchMaker trained on each task separately. The error bar represents 95% confidence interval of five-fold performance.

**Figure 6 molecules-28-00844-f006:**
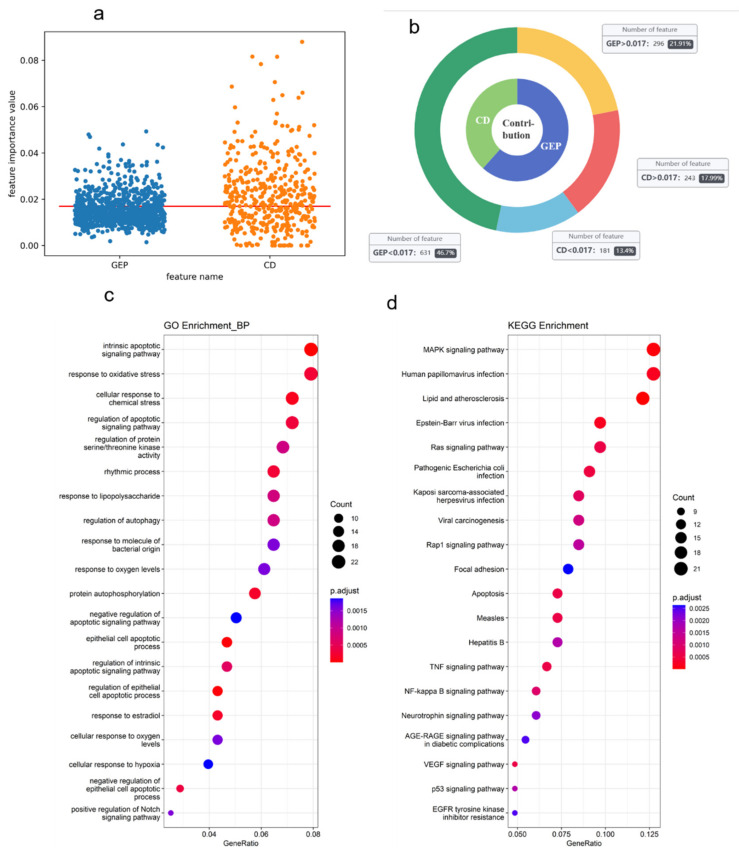
(**a**) The global importance values of gene expression profile (GEP) features and chemical descriptor (CP) features. The average importance value is 0.017. (**b**) The contribution of two types of features. The outer ring shows the number of features compared with the average importance value. (**c**,**d**) The top 20 enrichment results with (**c**) GOBPs and (**d**) KEGG pathway.

**Figure 7 molecules-28-00844-f007:**
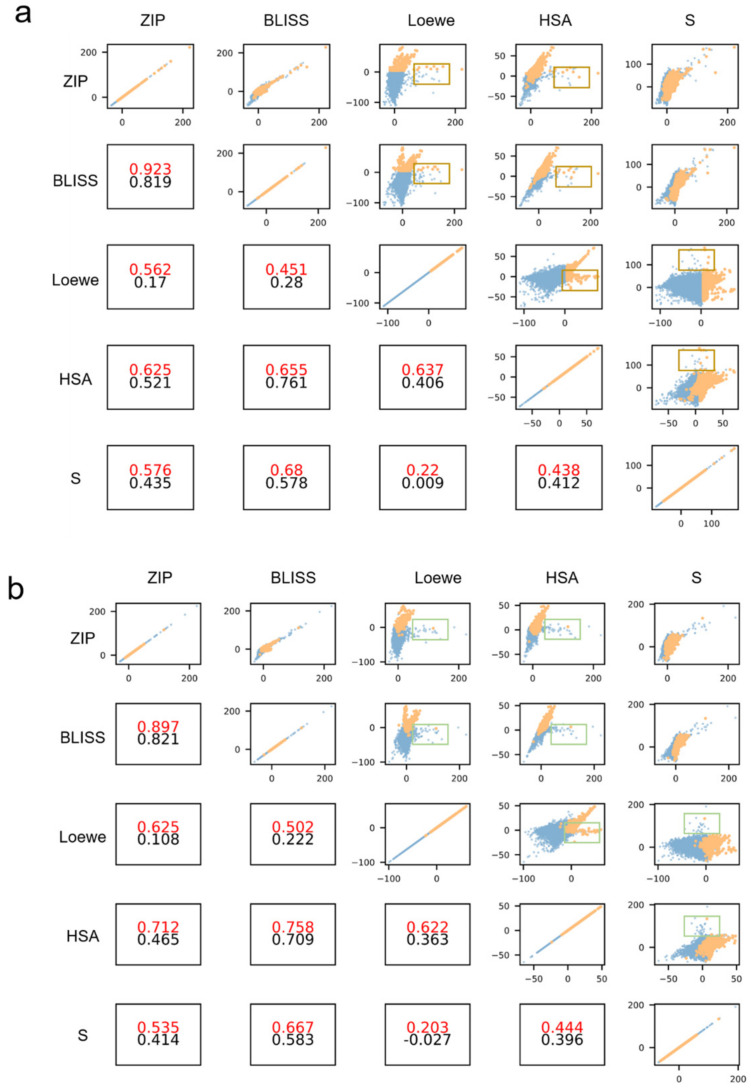
(**a**) The correlation between the different pairs of actual synergy scores. The yellow points represent the synergistic samples labeled by the actual Loewe score and the blue points represent the non-synergistic samples. (**b**) The correlation between the different pairs of predictive synergy scores. The yellow points represent the synergistic samples predicted by DEML.

**Figure 8 molecules-28-00844-f008:**
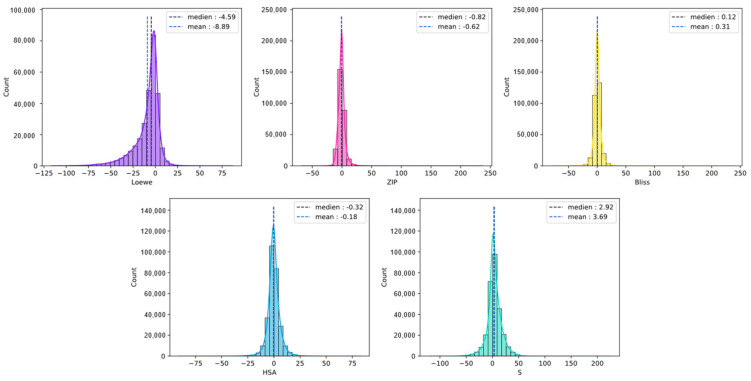
The distribution of different synergy scores in the dataset.

**Table 1 molecules-28-00844-t001:** Prediction performance of DEML tasks with different task transfer.

Model	Tasks Setting	AUROC	AUPR
DEML-onlyDDS	DDS	0.834	0.453
DEML-Loewe	Loewe, DDS	0.858	0.465
DEML-nonLoewe	ZIP, Bliss, HSA, S, DDS	0.872	0.482
DEML-REG	All synergy score, DDS	0.875	0.483
DEML-DDI	DDI, DDS	0.863	0.467
DEML	All task	0.894	0.526

**Table 2 molecules-28-00844-t002:** The expert ablation experiment results.

Experts Setting	RMSE	R2 Score	Spearman
DCE	9.159	0.616	0.740
BAE	8.694	0.648	0.758
BIE	8.469	0.666	0.777
DCE + BAE	8.472	0.661	0.774
DCE + BIE	8.690	0.651	0.762
BAE + BIE	8.469	0.667	0.774
All experts	8.319	0.679	0.792

**Table 3 molecules-28-00844-t003:** The top 10 key genes in four cancer cell lines.

HT29	MCF7	A549	SK-OV-3
STX4	IGF1R	SKIV2L	BZW2
SRC	PGM1	TSPAN14	TESK1
TUBB6	STX4	NCOA3	PAF1
GATA2	PHKG2	CCDC85B	CD97
WFS1	SPDEF	DNAJC15	RAB21
PCMT1	TSKU	IGF1R	STX4
KIT	PPP2R5E	PTGS2	FIS1
CCNH	RAD51C	SRC	GDPD5
TSKU	DDX42	TSKU	ZNF274
HEBP1	TBX2	WDTC1	TSKU

**Table 4 molecules-28-00844-t004:** The drug combinations with high predictive probabilities which are inconsistent with the actual Loewe score.

Drug1	Drug2	Cell Line	Pred ZIP	Pred BLISS	Pred HSA	Pred S	Pred Support	Pred Loewe	Actual Loewe	Actual ZIP	Actual Bliss	Actual HSA	Actual S	Support
ACTINOMYCIN D	ANTIBIOTIC AD 32	M14	6.84	7.7	8.48	20.53	5	5.66	−23.8	6.91	9.73	10.02	48.48	**** (+)
VEMURAFENIB	THALIDOMIDE	RPMI-8226	3.14	14.22	7.27	13.53	4	6.7	1.62	6.13	16.08	6.62	21.27	**** (+)
ACTINOMYCIN D	TAMOXIFEN CITRATE	K-562	23.76	28.61	22.52	45.81	5	18.2	0	23.17	23.69	24.28	39.77	**** (+)
DOCETAXEL	PLICAMYCIN	K-562	8.98	9.07	6.56	32.87	5	7.11	−0.69	11.98	11.93	7.51	18.84	**** (+)
VALACICLOVIR HCL	TEMOZOLOMIDE	T98G	23.09	22.33	12.72	25.43	5	12.24	4.77	13.77	13.77	4.77	15.67	*** (+)
PAZOPANIB HYDROCHLRIDE	VEMURAFENIB	RPMI-8226	13.08	20.64	11.48	16.77	5	7.02	0.15	7.23	14.25	2.22	16.65	*** (+)
NSC733504	ANTIBIOTIC AD 32	CCRF-CEM	13.67	13.9	12.17	51.76	5	13.78	1.17	7.74	12.45	1.95	17.86	*** (+)
GSK3787	TEMOZOLOMIDE	T98G	17.51	16.96	16.51	15.84	5	14.25	−3.39	10.9	10.9	−3.39	13.91	*** (+)
ZOLINZA	MK-2206	A2780	13.94	14.35	19.62	24.91	5	13.47	4.58	3.79	5.15	8.94	16.91	*** (+)
ANTIBIOTIC AY 22989	ANTIBIOTIC AD 32	EKVX	9.68	15.04	12.26	32.59	5	13.47	2.44	0.14	7.64	2.9	20.75	**(+)
ACTINOMYCIN D	TRETINOIN	OVCAR-8	10.45	12.67	16.71	27.18	5	7.03	−7.23	−5.05	−5.81	−3.64	19.39	*
GEFITINIB	LAPATINIB	UACC62	0.89	7.46	7.2	19.46	4	5.8	2.09	−3.35	4.33	3.61	12.76	* (+)
DOCETAXEL	SORAFENIB	OVCAR-8	11.81	13.36	8.58	21.59	5	9.01	2.54	1.92	3.83	3.11	−1.1	(+)
CHLORAMBUCIL	PLICAMYCIN	MDA-MB-468	6.03	6.52	8.94	16.45	5	8.98	2.39	2.05	2.91	2.45	−3.3	(+)
GNF-2	TEMOZOLOMIDE	T98G	26.18	25.92	26.19	25.66	5	28.03	−3.8	0.01	0.01	−3.8	0.82	
ARGATROBAN	TEMOZOLOMIDE	T98G	38.92	40.77	26.03	41.47	5	24.64	−4.85	−1.75	−1.75	−4.85	−1.1	
NSC733504	NSC256439	A498	12.22	7.36	16.34	17.05	5	14.85	1.22	−5	−5.24	1.83	−19.24	
ACTINOMYCIN D	VINBLASTINE SUFATE	NCI-H226	0.73	9.99	8.81	9.18	4	6.86	−11.11	−4.15	−5.57	−5.71	3.75	

Note: ‘pred support’ refers to the number of prediction synergy scores exceeding five. Support refers to the number of remaining actual synergy scores exceeding five. The number of “*” represents the number of the synergy scores with the certain drug combination higher than 5. “+” signal means the drug combination shows as synergistic in term of more than three ground truth synergy scores with a threshold of 0.

**Table 5 molecules-28-00844-t005:** The drug combinations with high predictive synergy scores in terms of all synergy score.

Drug1	Drug2	Cell Line	Pred Loewe	Pred ZIP	Pred BLISS	Pred HSA	Pred S	Probability	Actual Loewe	Actual ZIP	Actual Bliss	Actual HSA	Actual S	Support
ABIRATERONE	ACTINOMYCIN D	COLO 205	33.63	33.37	33.2	31.42	39.7	0.99	38.18	40.48	41.35	37.25	47.22	*****
ABIRATERONE	ACTINOMYCIN D	MOLT-4	37.12	38.28	40.03	38.88	45.88	0.97	46.32	45.13	48.17	47.45	54.19	*****
DACOMITINIB	TEMOZOLOMIDE	T98G	48.94	35.93	36.35	47.93	30.9	0.99	69.69	73.78	73.78	69.69	74.64	*****
PAZOPANIB HYDROCHLRIDE	ACTINOMYCIN D	MOLT-4	37.24	34.9	35.06	35.03	33.85	1	43.2	40.94	42.8	43.58	43.69	*****
REGORAFENIB (BAY 73-4506)	TEMOZOLOMIDE	T98G	31.51	34.13	33.73	32.31	32.55	0.99	37.96	32.82	32.82	37.96	31.74	*****
SALUBRINAL	TEMOZOLOMIDE	T98G	33.54	32.48	32.58	32.09	31.62	0.99	35.49	50.94	50.94	35.49	54.19	*****
TAK-733	TEMOZOLOMIDE	T98G	41.52	42.38	41.12	40.31	42.96	0.98	34.11	30.99	30.99	34.11	30.33	*****
VEMURAFENIB	ACTINOMYCIN D	HCT116	35.25	39.45	41.94	36.82	39.62	0.98	46.85	50.88	49.86	47.29	38.2	*****
VEMURAFENIB	ACTINOMYCIN D	MOLT-4	39.6	48.43	47.11	44.09	54.12	0.91	58.13	65.38	65.68	63.32	80.41	*****
VISMODEGIB	ACTINOMYCIN D	HCT116	33.48	33.95	36.96	33.96	34.65	0.98	38.26	39.19	41.01	38.89	34.03	*****
ENTINOSTAT (MS-275)	TEMOZOLOMIDE	T98G	48.48	51.04	50.59	48.35	52.75	0.99	19.36	7.45	7.45	19.39	2	****
TORIN 1	TEMOZOLOMIDE	T98G	45.13	40.71	38.79	44.65	43.72	0.98	9.61	7.03	7.03	9.69	−7.7	****
SNS-314 MESYLATE	TEMOZOLOMIDE	T98G	38.61	31.94	32.7	38.82	32.62	1	7.55	4.81	4.81	7.63	−9.76	**

Noted: The number of “*” represents the number of the synergy scores with the certain drug combination higher than 5.

**Table 6 molecules-28-00844-t006:** The drug combinations predicted with synergistic and adverse DDIs. DDI probability refers to the DEML predictive probability in DDI classification tasks.

Drug1	Drug2	Cell line	Support	Actual ZIP	Actual BLISS	Actual HSA	Actual S	Actual S	DDI Probability
DOCETAXEL	LAPATINIB	NCIH23	*****	19.1	23.4	21.14	21.44	26.93	0.88
DOCETAXEL	SUNITINIB	NCI-H322M	****	5.17	6.8	10.53	8.22	3.85	0.93
DOCETAXEL	SUNITINIB	M14	*****	7.36	10.17	8.03	8.29	22.04	0.94
DOCETAXEL	PLICAMYCIN	K-562	*****	11.98	11.93	−0.69	7.51	18.84	0.91
DOCETAXEL	SORAFENIB	COLO 205	*****	20.33	19.1	12.76	12.99	43.45	0.97
DOCETAXEL	SORAFENIB	OVCAR-8	*****	1.92	3.83	2.54	3.11	−1.1	0.96
DOCETAXEL	SORAFENIB	SK-MEL-28	*****	9.26	10.88	12.33	12.61	21.41	0.98
ERLOTINIB	DASATINIB	LOVO	**	−6.09	−5.84	7.83	9.35	−48.67	0.94
IMATINIB	DOCETAXEL	COLO 205	*****	15.34	15.68	15.6	16.62	33.16	0.96
ABIRATERONE	5-FU	NCI-H322M	*****	5.51	9.47	7.11	8.95	15.99	0.93
ABIRATERONE	LOMUSTINE	UO-31	*****	−0.91	4.75	5.93	6.66	3.12	0.94
ABIRATERONE	PACLITAXEL	RXF 393	*****	10.1	10.1	10.94	12.71	31.99	0.94
DOCETAXEL	LAPATINIB	NCIH23	*****	19.1	23.4	21.14	21.44	26.93	0.88

Noted: The number of “*” represents the number of the synergy scores with the certain drug combination higher than 5.

**Table 7 molecules-28-00844-t007:** Classification label data details.

Classification Task	Numbers of Positive Samples	Numbers of Negative Samples	Criteria of Label Definition
Synergy classification task	18,060	268,361	Labeled by Loewe score using threshold 5
DDI classification task	20,955	265,466	Labeled by DrugBank descriptions

**Table 8 molecules-28-00844-t008:** The drug combinations with high predictive synergy scores in terms of all synergy scores.

Feature Name	Number of Features	Feature Dimension	Numerical Types of Features
Chemical descriptor	3040	541	Continuous
Cell line gene expression profiles	81	927	Mix of discrete and continuous

## Data Availability

Both the code developed in this research and the related datasets are available at the following Github repository: https://github.com/saya34tju/DEML, accessed on 1 December 2022.

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
