# Peer review of "DEML: Drug Synergy and Interaction Prediction Using Ensemble-Based Multi-Task Learning"

_molecules, 2023, doi:10.3390/molecules28020844_

Round 1
Reviewer 1 Report
1. The overall grammar and format of the article is very well, but there are still some minor faults with it, such as the period in line 194 and the "Ok" in line 583.
2. The article divides the data set in such a way that the test set is not separated from the training set. Although the early-stopping is adapted, it may still affect the generalization of the method. Please give reasons for doing so and evidence that this does not affect the generalization ability of the model.
3. There are also some problems with the formulas in the article: the "n" in Equation 4 does not seem to have a precise meaning; the "I" in Equation 10 is not explained in the article; and the "m" in Equations 17 and 18 is not reflected in the article.
4. The article takes a bitwise multiplication of the two vectors in the Bi-interaction expert section but does not explain it in detail; you should clarify the reasons for doing this and the positive impact it has on the overall model.
Author Response
Please see the attachment.
The attachment contains the response content and revision manuscript.
Using the "Track Changes" function with MS Word can viewed the change about the manuscript.

Reviewer 2 Report
This paper models five metrics of drug-drug synergy using multi-task and ensemble methods. Overall, this looks interesting and should be published in some form.
Scientific questions:
The DEML architecture looks quite complex, and some more explanation is needed. On the left of Figure 1 are three architectures that seem to have three neurons in the output layer. Are these architectures the share-bottom, SNR-trans, and SNR-aver architectures, or something else? Does the left block represent three architectures applied to a single task (e.g. Bliss) or is each architecture "multitask" (in which case there should be 5 output neurons)? Do the architectures with two separate input blocks represent input from two molecules?
Presumably the entities being modeled are pairs of molecules, at least for the single input block architecture. How does one combine the gene data (perhaps by correlation between profiles?) and chemical features from individual molecules?
One difficulty with these architectures is that they are uninterpretable without something like SHAP analysis. However, random forest could get feature importances. It would be useful to know what the determinations synergy are, e.g. (to make something up) "if the gene profiles of two drugs are correlated, and they are both lipophilic, synergy is likely to be high."
Random forest might look better in the categorical models if you used in "balanced mode".
Editorial:
It would be useful to explain how synergy is defined and what the individual indices (e.g. Bliss) mean and what the range of values is. Is one index more easily predicted and therefore might be more internally consistent?
Why report both Pearson and R2 since these track perfectly? Similar with MSE and RMSE. AUPR seems to track nearly perfectly with AUROC also. It is the usual convention to use R2, RMSE, and AUROC.
Figure 1 "hybrid" is misspelled
Author Response

(The authors gave the same response as above.)

Round 2
Reviewer 2 Report
This paper regression models five metrics (plus categorical models two labels) of drug-drug synergy using multi-task and ensemble methods. Overall, this looks interesting and should be published in some form. The analysis is thorough and now contains a discussion of feature importance. The dissection of the effects of different features and architectures is also very good. I believe the paper demonstrated that the DEML approach is better than alternative approaches. My major issue for the first version has been partly addressed, in the sense that Figure 1 is much better notated. However, I still have some difficulty understanding the architecture, and I hope the authors rethink how this is introduced. Remaining issues in Figure 1: There are three output nodes in the neural nets that I do not understand. Also, what is the origin of tower "B2"?
The only scientific issues to raise:
1. the classical method of "early stopping" is used to reduce overfitting instead of the more modern "dropout". Is there any reason for that?
2. Also, typically for regression models, one typically uses R2, the square of "Pearson".
Most of my comments are editorial in there seems to be a number of grammatical errors. For example, here are some suggested changes in language:
Line 113 "The multiple vector input to the task-specific fusion layer." is not a sentence
Line 122 "different types of expert subnetwork."
Line 161 Do not understand "This is likely because ML-based model has limitations on large scale datasets." If this implies that non-deep-neural-net methods have issues with very large datasets, that is probably not true.
Line 236 "then fuse the information"
Line 293 "This result is consistent with the high correlation between HSA score and Loewe Score"
Line 308 "and puts the extraction representation"
Line 433 "shows higher consistency"
Line 582 "as strong as expected"
Line 591 "Then we analyze the correlation of different synergy"
Line 649 "automatically convert input"
